# Predicting Cardiac Arrest and Respiratory Failure Using Feasible Artificial Intelligence with Simple Trajectories of Patient Data

**DOI:** 10.3390/jcm8091336

**Published:** 2019-08-29

**Authors:** Jeongmin Kim, Myunghun Chae, Hyuk-Jae Chang, Young-Ah Kim, Eunjeong Park

**Affiliations:** 1Department of Anesthesiology and Pain Medicine, Severance Hospital, Yonsei University College of Medicine, Yonsei University Health System, Seoul 03722, Korea; 2Computer Engineering, AI R&D Lab. of Selvas AI Inc., Seoul 08594, Korea; 3Division of Cardiology, Severance Cardiovascular Hospital, Yonsei University College of Medicine, Yonsei University Health System, Seoul 03722, Korea; 4Division of Medical Information and Technology, Yonsei University Health System, Seoul 03722, Korea; 5Cardiovascular Research Institute, College of Medicine, Yonsei University, Seoul 03722, Korea

**Keywords:** deep learning, cardiac arrest, respiratory failure, intensive care unit, machine learning, artificial intelligence

## Abstract

We introduce a Feasible Artificial Intelligence with Simple Trajectories for Predicting Adverse Catastrophic Events (FAST-PACE) solution for preparing immediate intervention in emergency situations. FAST-PACE utilizes a concise set of collected features to construct an artificial intelligence model that predicts the onset of cardiac arrest or acute respiratory failure from 1 h to 6 h prior to its occurrence. Data from the trajectory of 29,181 patients in intensive care units of two hospitals includes periodic vital signs, a history of treatment, current health status, and recent surgery. It excludes the results of laboratory data to construct a feasible application in wards, out-hospital emergency care, emergency transport, or other clinical situations where instant medical decisions are required with restricted patient data. These results are superior to previous warning scores including the Modified Early Warning Score (MEWS) and the National Early Warning Score (NEWS). The primary outcome was the feasibility of an artificial intelligence (AI) model predicting adverse events 1 h to 6 h prior to occurrence without lab data; the area under the receiver operating characteristic curve of this model was 0.886 for cardiac arrest and 0.869 for respiratory failure 6 h before occurrence. The secondary outcome was the superior prediction performance to MEWS (net reclassification improvement of 0.507 for predicting cardiac arrest and 0.341 for predicting respiratory failure) and NEWS (net reclassification improvement of 0.412 for predicting cardiac arrest and 0.215 for predicting respiratory failure) 6 h before occurrence. This study suggests that AI consisting of simple vital signs and a brief interview could predict a cardiac arrest or acute respiratory failure 6 h earlier.

## 1. Introduction

Unexpected cardiac arrest or acute respiratory failure requires immediate attention as these are critical emergent events that often cause catastrophic repercussions, including death or medicolegal issues, if not treated in a timely manner. If a person suffering from cardiac arrest is not recovered by spontaneous circulation, the organs and tissues of the body will not receive enough blood, and the cells will not be supplied with oxygen and nutrients, resulting in organ failure and death. With regard to respiratory failure, one of the most critical tasks for a physician managing an acutely unstable patient is to secure the patient’s airway. The inability to oxygenate despite non-invasive oxygen therapy may lead to an immediate life threat and, with rare exceptions, mandates endotracheal intubation. Therefore, predicting the timing of tracheal intubation is very useful. However, there is no guideline to predict adverse events to manage all possible scenarios requiring preemptive care.

An Early Warning Score (EWS) based on vital signs is a medical service that can be used for screening deteriorating patients to quickly determine their degree of illness. The Modified Early Warning Score (MEWS) and the National Early Warning Scores (NEWS and NEWS2) are the most widely used scoring systems [1,2]. Each component of the scoring system includes basic physiologic data, such as blood pressure and heart rate. A higher score in this tool is statistically related to an increase in the probability of an adverse outcome. In the general ward, EWS is often used as part of a “track and trigger” system; a high score is linked to the medical emergency team or rapid response team reviewing the patient’s status so that a critical intervention is not delayed.

Recently, artificial intelligence (AI) has been introduced in the clinical field, and studies have been published to predict adverse events (including sudden cardiac arrest) several hours before their occurrence and have found that the deep learning model is more useful for detecting high-risk patients compared with the existing EWS [3,4,5,6,7]. Most AI models leverage massive and complicated data, including the various monitoring parameters and laboratory data of electronic medical records (EMRs) collected in hospital information systems [3,6,7,8,9]. Recent studies demonstrate that deep learning, especially associated with recurrent neural networks (RNNs) utilizing long short-term memory (LSTM) or gated recurrent units (GRUs) to predict events using time-series data, outperforms logistic regression based predictions or clinical scores. The reason for this finding is that LSTM or GRUs can suitably control the influence of previous input data with multiple features in every training stage [10,11,12]. These models combine sequential features, including multiple values from laboratory tests or medical devices, and nonsequential features, such as diagnoses or procedures, in data from the intensive care unit (ICU).

However, to date, a procedure for the selection of optimal feature sets for various machine learning algorithms has not been determined [13,14]. There are several considerations to be taken into account for determining the features to be included in a feasible predictive model for acute adverse events. First, they should be reproducible and verifiable in a variety of clinical settings, and common medical data should be available. Second, when interpreting physiologic data, clinical intervention should be considered. For example, if a patient with a mean blood pressure (MBP) of 40 mmHg is administered a vasopressor, the MBP value should be weighted over the untreated patient’s value. Third, it is more realistic to exclude laboratory data that are prone to time delays. Blood tests are not available in all medical situations, and it takes approximately one hour after blood collection to obtain the laboratory tests.

In this paper, we used six major physiological features and three major demographic features, including the history of recent surgery, current health status and treatment history, to create Feasible Artificial intelligence with Simple Trajectories for Predicting Adverse Catastrophic Events (FAST-PACE), a neural network model that can predict the occurrence of adverse events (at 1 h, 2 h, 4 h, and 6 h) in a real-world setting. In this retrospective study, which includes adverse event cases, we assess the merit of a predictive machine learning approach to increase the quality of care and patient safety. Although various monitoring devices in the ICU are constantly producing data, we endeavored to construct a deep learning model using only physiologic features, owing to their reliability, reproducibility, and real-time measurability.

## 2. Methods

### 2.1. Study Population and Data Sources

A total of 29,181 ICU patients were included from two hospitals within the Yonsei Health System (Severance Hospital and Gangnam Severance Hospital) from 2006 to 2017. The study protocol was approved by the Institutional Review Board of the Yonsei University Health System, Seoul, South Korea (Sinchon Severance Hospital and Gangnam Severance Hospital (#4-2017-1230 and #3-2018-0263, respectively)). We included patients above the age of 18 who were hospitalized in one medical ICU, one surgical ICU, and one mixed ICU with a total of 67 ICU beds. Patients were excluded if they underwent events within one hour of ICU admission or if they did not survive the first 6 h in ICU (1231 patients underwent acute respiratory failure events and 242 patients underwent cardiac arrest events; see Figure 1). In this model, we define cardiac arrest as the start of cardiopulmonary resuscitation (CPR), and acute respiratory failure as endotracheal intubation. Both events were detected and managed by the rapid response system conducted in ICU. More specifically, as it is invasive, endotracheal intubation can result in many complications and is a rather difficult clinical procedure. Physicians attempt endotracheal intubation as a last resort; that is, when respiratory failure cannot be resolved with the provision of any other non-invasive oxygen supply. We define this point as respiratory failure requiring immediate intervention. In real-world applications of learning methods in medicine, the ratio of positive to negative instances is significantly low, and such imbalanced classes restrict prediction performance, although machine learning is a promising solution when using big data in medical machine learning [15,16]. As shown in Figure 1, FAST-PACE also suffered from intrinsic imbalanced data. Therefore, we performed random undersampling, in which the training dataset was modified to produce a less balanced class distribution to allow learning to be conducted as a standard prediction. We randomly selected 3984 samples for the training dataset, which was composed of an identical number of positive (event group: 1992 samples) and negative (no-event group: 1992 samples) samples [17]. However, we maintained the imbalanced ratio of positive to negative samples to demonstrate that the proposed solution is feasible in a real-world environment without overfitting, as shown in the FAST-PACE test of Figure 1.

### 2.2. Feature Construction

In this study, we retrieved a total of nine features from EMRs to predict two critical events—acute respiratory failure and cardiac arrest. The features used to develop this model were basic vital signs, such as pulse rate, systolic blood pressure (SBP), diastolic blood pressure (DBP), respiratory rate, peripheral oxygen saturation (SpO_2_), body temperature, recent surgical history (within one week), and current health status (American Society of Anesthesiologists (ASA) classification). The prediction times are taken as 1 h, 2 h, 4 h, and 6 h to analyze the tendency of an adverse event, depending on the prediction time window (*P_tw_* = 1, 2, 4, and 6). To predict the probability of a critical event occurring at a specific time, data recorded from admission to *P_tw_* hours before the event can be used for the model construction. Therefore, as shown in Figure 2, the EMR trajectories up to *P_tw_* hours before the reference point were retrieved to build the feature set of FAST-PACE.

Pulse rate, SBP, DBP, respiratory rate, SpO_2_, and body temperature were extracted for the investigation of vital signs of patients, and three additional information parameters (treatment history, current health status, and recent history of operations) were encoded. The treatment history was determined as a binary feature of pharmacological treatment and additional oxygen supply that could affect the vital signs at the time of measurement. The included drugs are vasopressors and inotropics, such as norepinephrine, vasopressin, phenylephrine, epinephrine, dobutamine, and dopamine. The oxygen supply includes all additional oxygen supply from the low flow system to the extracorporeal membrane oxygenation and is indicated as a 0 or 1 as a binary indicator. The recent surgery history was defined within one week of event occurrence. A few patients were also assessed by two conventional risk scores—MEWS and NEWS.

EMR data from ICUs contain abnormal records due to errors of the medical staff or unexpected omissions of data during the input process. We adopted the imputation and discretization methods to deal with such noise, missing values, and various ranges of features. Initial data cleaning was performed to refine the mixed value of each physiological signal as well as to remove outliers or invalid data since the initial vital sign values were recorded in a specific range rather than as a single value, or included additional information such as history of treatment, major symptoms, and units. The range value was replaced with the lowest value, and additional information including treatment history were separated to other features. Missing data were imputed by applying autoregressive integrated moving average (ARIMA) models, which have been widely used for time-series imputation [18].

We define treatment history as medication administration or oxygen supplements that can affect heart rate, SpO_2_, SBP, DBP, and temperature using medication history and nursing records. The features were sampled hourly and discretized using the mean and standard deviation from the training set.

The ASA physical status classification system assesses the fitness of patients before surgery. In 1963, the ASA adopted the five-category physical status classification system; a sixth category was added later. This system, which continues to be a means of stratifying a patient’s systemic illness [19,20], is defined as follows:
Class 1. Healthy person.Class 2. Mild systemic disease.Class 3. Severe systemic disease.Class 4. Severe systemic disease that is a constant threat to life.Class 5. A moribund person who is not expected to survive without an operation.Class 6. A declared brain-dead person whose organs are being removed for donor purposes.

Because ASA records exist only for patients who underwent surgery recently, the value was only extracted for 10% of the patients. The system provided by the ASA is a tool for evaluating the severity of underlying disease in patients using a few simple questions. Researchers, however, have not been able to extract ASA data for non-surgical patients. In future studies, the ASA questionnaire could be revamped to include a few simple questions to the above effect. Therefore, we included ASA data as a continuous feature in the model. The resulting z-scores were rounded to be within a range, differing depending on the features, as shown in Table 1. Sequential repeated measurements of vital signs, pulse rate, SBP, DBP, SpO_2_, and temperature were supplemented with missing values using an ARIMA model.

To compare the prediction performance of FAST-PACE to the baseline, we also obtained the values of MEWS and NEWS, which are conventional scores for the assessment of patients in wards. The composition of the scores is displayed in Table 2.

### 2.3. Deep Learning Model

Vital signs data of ICU patients are a time series of periodic records. Recent studies have applied RNNs to analyze and predict patterns of a patient’s condition [3,22,23]. Specifically, RNNs using the LSTM model equipped with memory cells to store trajectory information is broadly adopted in diagnosis and prediction in healthcare [24,25]. At each time step, LSTM reads an input *x_i_*, updates the memory cell (*S_i_*), and returns an output, as shown in Figure 3. Each input *x_i_* is a two-dimensional vector composed of nine features and a time window, and the final output p denotes the probability of the occurrence of a critical event. LSTM extends memory block-typed neurons so that memory cells in nodes can properly control the influence of the previous input. In this study, LSTM is composed of a single hidden layer with 128 cells and a drop-out probability of 0.5 was applied as a normalization technique [26]. LSTM equations with time-series inputs are detailed in [10]. In addition, the *Adam* (adaptive moment) optimization technique was applied to enhance the prediction performance [27].

To generalize the proposed solution, the initial weights were set via Xavier initialization [28]. Furthermore, early stopping was used, whereby training was stopped at the lowest error achieved on the validation set to improve generalization [29]. We trained the model in TensorFlow 1.6 [30] with the Python 3.5, pandas 0.19, NumPy 1.12, and SciPy 1.01 libraries.

## 3. Results

In this section, we present the experimental results to evaluate the feasibility of FAST-PACE with simple trajectory and improvement in prediction performance. Table 3 lists the baseline characteristics of the patients. To validate our model, we also investigated MEWS and NEWS, which are based on the AVPU (alert, verbal, pain, unresponsive) scale and are broadly adopted to measure the severity of patient condition. The MEWS and NEWS records of 20,436 patients are also included in Table 3.

First, we investigated the event distribution of acute respiratory failure and cardiac arrest over time after admission to the ICU in order to demonstrate the importance of predicting adverse events in the 1–6 h after admission. Figure 4 shows a histogram of the 29,181 ICU patients described in Section 2. As seen in the figure, the number of events occurring 1–5 h after admission is relatively high and gradually decreases henceforth.

The prediction performances of FAST-PACE, MEWS, and NEWS were compared in terms of area under the receiver operating characteristic curve (AUROC), sensitivity, specificity, positive predictive value (PPV), negative predictive value (NPV), accuracy, and F2-score. The cutoff value for the occurrence of events was 5 for NEWS and MEWS, and 0.5 for the LSTM model.

Table 4 shows the performance of MEWS, NEWS, and FAST-PACE for predicting acute respiratory failure 1 h, 2 h, 4 h, and 6 h before the catastrophic event. FAST-PACE predicts acute respiratory failure with an AUROC of 0.868–0.886, while the AUROC values for MEWS and NEWS are 0.607–0.634 and 0.608–0.641, respectively. FAST-PACE improved the prediction in terms of AUROC compared to MEWS and NEWS by over 40% on average (0.877, 0.620, and 0.623, respectively). Above all, prediction with FAST-PACE outperforms other traditional warning scores by reaching a mean sensitivity of 0.830 (0.771–0.881), which is an increase from a mean sensitivity of 0.222 (0.201–0.245) for MEWS and 0.491 (0.467–0.518) for NEWS, while in terms of specificity, MEWS performed slightly better than FAST-PACE and NEWS (0.876, 0.767, and 0.705 on average, respectively).

Table 5 lists the performance of MEWS, NEWS, and FAST-PACE in predicting cardiac arrest 1 h, 2 h, 4 h, and 6 h before the critical event. FAST-PACE improved the AUROC of prediction to 0.896 for the 1 h time window, while the AUROC values for MEWS and NEWS were 0.746 and 0.759. The average sensitivity of FAST-PACE was 0.844 (0.814–0.870), which improved the average sensitivity of MEWS (0.400) and NEWS (0.695). FAST-PACE enhanced the overall performance compared to NEWS in terms of AUROC, sensitivity, and accuracy, while it approached that of MEWS for specificity (0.767–0.876). To compare FAST-PACE and other conventional scoring systems, all p values at each point were statistically significant by McNemar test (*p* < 0.001).

Figure 5 shows the ROC of FAST-PACE, MEWS, and NEWS, predicting critical events within 6 h. We also measured net reclassification index (NRI) to evaluate the improvement in the prediction performance over conventional warning scores, as shown in Table 6 and Table 7. NRI from MEWS to FAST-PACE is 0.341 for predicting respiratory failure and 0.507 for predicting cardiac arrest, and NRI from NEWS to FAST-PACE is 0.215 for predicting respiratory failure and 0.412 for predicting cardiac arrest 6 h before event occurrence. Although the NRI from MEWS to FAST-PACE in classifying the no-event group is slightly negative due to MEWS’ overfitted specificity, the gain from improved prediction performance was the highest in predicting cardiac arrest 2 h prior to occurrence.

## 4. Discussion

In this paper, we have developed a deep learning model, FAST-PACE, that predicts acute cardiac arrest and respiratory failure at different time intervals with a simple clinical trait. This study demonstrated two significant results: an applicable AI model in a clinical environment and improved prediction performance. The proposed system aims to capitalize on the prediction capability of AI with simple trajectories and is feasible to use for instant decision support in wards or pre-hospital environments. We aim to maintain the feature set to be as concise as possible while achieving reasonable performance in order to make FAST-PACE applicable to any environment where instant prediction is necessary. In this respect, we fed a simple trajectory of patient data to the learning model that can increase the feasibility of application, which is different to other studies that utilized big data with a significant number of features. Although FAST-PACE was trained with the hemodynamic parameters commonly used in EWS and the recent history of operations, it achieves higher performance in AUROC (0.886 ± 0.010) than MEWS (0.737 ± 0.012) or NEWS (0.750 ± 0.014) for prediction 1–6 h before acute cardiac arrest. In particular, the performance of FAST-PACE was demonstrated for prediction sensitivity; MEWS and NEWS performed with unacceptable sensitivity (0.388 ± 0.019 and 0.685 ± 0.022) with simple trajectories, while FAST-PACE achieved 0.857 (±0.045) in sensitivity. Such enhancement in sensitivity of deep learning-based prediction over EWS was investigated in predicting both acute respiratory failure and cardiac arrest, as displayed in Table 4 and Table 5. The primary reason for this performance difference is that FAST-PACE reflected the effect of current treatment on each physiological parameter by adding treatment information as binary factors and weighed on the simplest past medical history that can be gained through short interviews. In a previous report, Lee et al. reported that deep neural networks can be utilized to predict in-hospital mortality based on automatically extractable and objective intraoperative data. It has been demonstrated that these predictions are further improved via the addition of preoperative information, such as ASA score, to predict postoperative mortality [20]. In order to predict the adverse events after major surgery, it is necessary to discriminate whether the postoperative adverse event is due to the major operation itself or the patients’ preoperative medical condition. The authors assume that this would have made a difference in FAST-PACE.

Another advantage of applying FAST-PACE is that previous ICU-based AI studies have utilized prediction models with as many features as possible, whereas we constructed a model without any laboratory data or echocardiography data with a view of providing a feasible solution to various environments, including general wards, ICUs, and pre-hospital emergency environments. Most deep learning models incorporate not only laboratory data, but also use echocardiograms [6,31]. A previous study used 37 clinical risk factors to predict failed extubation [9]. Such deep learning models require several healthcare features, including both static features such as general descriptors (demographic information collected during admission), and temporal features, which possibly come from various monitoring devices, injury markers, ventilator settings, blood gas values, and other sources [8,9]. However, the medical data acquired during cardiopulmonary resuscitation or ambulance transfer are extremely limited. For example, the collection of 12-lead electrocardiogram data cannot be expected in decision support using a prediction model. This is why we need to develop a model that predicts emergency situations by analyzing limited real-time physiological data. Although we can obtain huge amounts of data from various monitors in the ICU, we have built models that include only physiological data and past surgical history, excluding laboratory data. Our models can be easily applied to a limited population, such as patients in ambulance transfer or general ward patients.

Predictive scoring systems are typically used to predict mortality in general ICU patients. The acute physiologic and chronic health evaluation (APACHE) [32], simplified acute physiologic score (SAPS) [33], sequential organ failure assessment (SOFA) [34], and mortality prediction model (MPM) [35] are the four major validated ICU predictive scoring systems. These scoring systems are useful for determining prognosis for the patient or a severity index in the ICU. However, since a patient’s condition can continuously change, these scoring tools are unable to predict, in real-time, catastrophic events such as sudden cardiac arrest or acute respiratory failure. In addition, all ICU scoring except MPM is measured after 24 h in the ICU. As shown in Figure 4, most adverse events occur within 24 h of admission to the ICU. With these scoring systems, it is difficult for physicians to predict critical events or make prognoses.

Physicians expect to spend less time analyzing medical data using AI and to quickly screen high-risk patients. However, respondents pointed out that AI may not be able to help unexpected situations due to inappropriate data [36]. By applying this model using simple traits, we have the time to plan for a patient to be treated at a medical facility with sufficient medical personnel. For example, when we transfer a critical patient by ambulance, we can use our system to automatically notify the nearest local emergency center of the patient’s condition and request advance preparation before the patient arrives.

Although the proposed solution demonstrates enhanced performance in predicting catastrophic events in mixed ICU populations, we need to apply this system to external data or a progressive study to validate the performance of the trained model. In addition, the results suffer from the limitation that all data were retrieved from ICU patients and none were associated with patients from other environments, such as emergencies and on-board predictions in ambulances. Thus, the expansion of the data to include medically urgent situations remains an important topic for future research. Research adopting various technical methods to treat imbalanced classes is also necessary. We plan to apply the synthetic minority over-sampling technique (SMOTE) [37], oversampling, or cost-sensitive methods [38] with various weightings for classes to solve the imbalance problem, which is common in medical big data. In addition, our data set is based on data collected in one-hour intervals. In recent years, many medical institutions monitor a patient’s vital signs in real time and store and display the data on the minute. Since the patient’s condition may change rapidly, we plan to build a model based on vital signs collected in real time.

## 5. Conclusions

In this study, we developed a deep learning-based prediction model for acute emergency situations requiring immediate intervention. We used AI for instant prediction of the occurrence of adverse events with limited physiologic traits and simple past medical history. Only simple clinical traits obtained from 1 h to 6 h prior to adverse events were utilized to accurately predict acute cardiac arrest or respiratory failure. This suggests that a monitoring alert system and life-saving strategy can be implemented shortly before an adverse event.

## Figures and Tables

**Figure 1 jcm-08-01336-f001:**
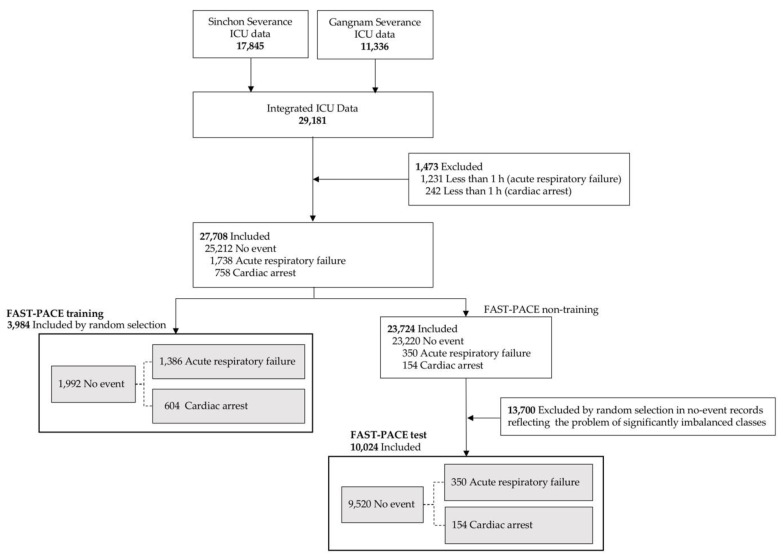
Studied populations. ICU = intensive care unit; FAST-PACE = Feasible Artificial intelligence with Simple Trajectories for Predicting Adverse Catastrophic Events.

**Figure 2 jcm-08-01336-f002:**
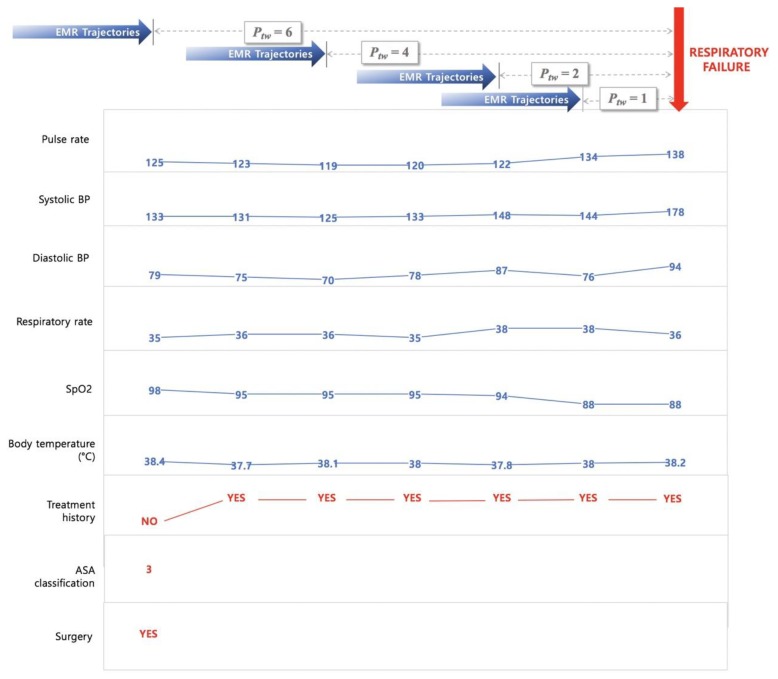
Sample trajectories of patient with acute respiratory failure. *P_tw_* = prediction time window; ASA = American Society of Anesthesiologists; EMR = electronic medical record; BP = blood pressure; SpO_2_ = peripheral oxygen saturation.

**Figure 3 jcm-08-01336-f003:**
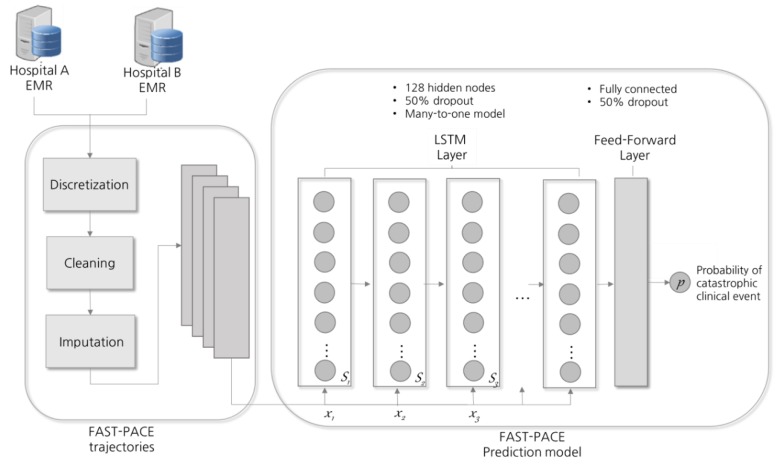
Prediction model design. LSTM = long short-term memory; *x* = input; *S* = memory cell.

**Figure 4 jcm-08-01336-f004:**
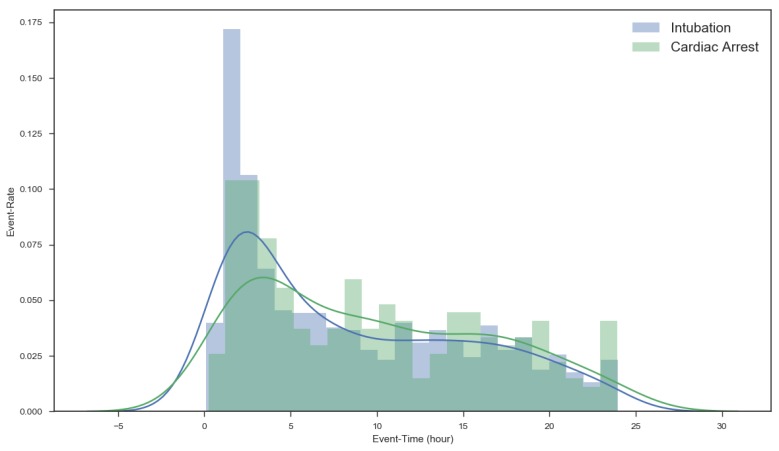
Event distribution after admission.

**Figure 5 jcm-08-01336-f005:**
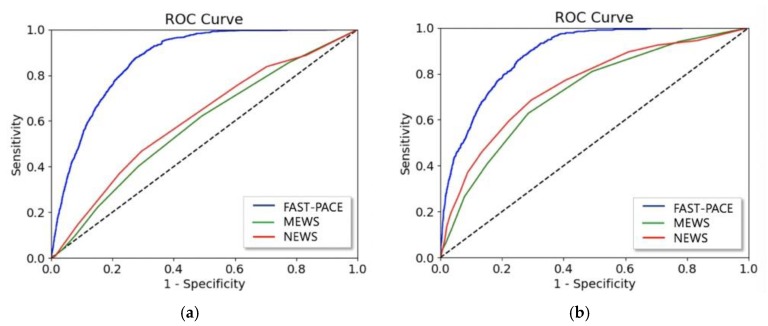
AUROC of FAST-PACE, MEWS, and NEWS predicting (**a**) acute respiratory failure and (**b**) cardiac arrest within 6 h.

**Table 1 jcm-08-01336-t001:** List of features.

Category	Feature	Data Type	Range	Missing (%)
Vital	Pulse rate (bpm)	continuous	0–300	11.46
Sign	Systolic BP (mmHg)	continuous	0–300	7.78
	Diastolic BP (mmHg)	continuous	0–300	6.81
	Respiratory rate (breaths/min)	continuous	0–150	12.76
	SpO_2_ (%)	continuous	0–100	24.01
	Body temperature (°C)	continuous	2–45	14.36
History	Treatment history ^†^ (yes or no)	categorical	0, 1	
Operation ^‡^	ASA classification	continuous	1–6	
	History of recent surgery (yes or no)	categorical	0, 1	

^†^ Treatment history: any pharmacological treatment or additional oxygen supply that could affect the vital signs at the time of measurement; ^‡^ Operation: major surgery within one week of event occurrence.

**Table 2 jcm-08-01336-t002:** Modified Early Warning Score (MEWS) and National Early Warning Score (NEWS) [2,21].

MEWS	3	2	1	0	1	2	3
Respiratory rate (breaths/min)	>35	31–35	21–30	9–20			<7
SpO_2_ (%)	<85	85–89	90–92	>92			
Temperature (°C)		>38.9	38–38.9	36–37.9	35–35.9	34–34.9	<34
Systolic BP (mmHg)		>199		100–199	80–99	70–79	<70
Heart rate (bpm)	>129	110–129	100–109	50–99	40–49	30–39	<30
AVPU ^†^				Alert	Verbal	Pain	Unresponsive
NEWS	3	2	1	0	1	2	3
Respiratory rate (breaths/min)	≥25	21–24		12–20			≤8
SpO_2_ (%)	≤91	92–93	94–95	≥96			
Temperature (°C)		≥39.1	38.1–39	36.1–38.0	35.1–36		≤35
Systolic BP (mmHg)	≥220			111–219	101–110	91–100	≤90
Heart rate (bpm)	≥131	111–130	91–110	51–90	41–50		≤40
AVPU ^†^				Alert			Verbal, pain, Unresponsive

^†^ AVPU (alert, verbal, pain, unresponsive) is a system by which a health care professional can measure and record a patient’s level of consciousness.

**Table 3 jcm-08-01336-t003:** Patient demographics.

Feature	FAST-PACE Training			FAST-PACE Test			MEWS, NEWS Score		
	Acute Respiratory Failure (*n* = 1388)	Cardiac Arrest (*n* = 604)	Non-Event (*n* = 1992)	Acute Respiratory Failure (*n* = 350)	Cardiac Arrest (*n* = 154)	Non-Event (*n* = 9520)	Acute Respiratory Failure (*n* = 746)	Cardiac Arrest (*n* = 102)	Non-Event (*n* = 19,903)
Age (years)	62.2 ± 15.7	62.9 ± 15.4	62.9 ± 15.3	62.1 ± 14.9	63.6 ± 14.4	61.5 ± 15.5	63.9 ± 15.5	64.3 ± 14.1	61.7 ± 15.8
Gender (male), *n* (%)	842 (60.6)	382 (63.2)	1225 (61.4)	225 (64.2)	106 (68.8)	5805 (60.9)	451 (60.4)	73 (71.5)	12,001 (60.2)
Race, Asian	1388	604	1992	350	154	9520	746	102	19,903
Pulse rate, (bpm)	100.7 ± 22.2	107.4 ± 24.4	97.3 ± 23.3	100.1 ± 20.7	108.5 ± 25.3	91.0 ± 21.4	99.3 ± 21.8	103.9 ± 23.1	89.8 ± 20.9
Systolic BP (mmHg)	127.6 ± 24.5	110.3 ± 26.1	125.6 ± 24.2	126.9 ± 23.3	107.8 ± 26.2	126.4 ± 26.	127.7 ± 23.8	110.6 ± 27.7	127.4 ± 25
Diastolic BP (mmHg)	67.5 ± 14.4	59.2 ± 15	66.7 ± 14.3	66.9 ± 13.3	58.5 ± 14.8	66.7 ± 13.9	67.3 ± 14	58.8 ± 14.5	67.2 ± 13.7
Respiratory Rate (breaths/min)	22.8 ± 6.8	22.5 ± 6.3	21.4 ± 6.4	23.4 ± 7.	22.9 ± 6.3	18.6 ± 5.3	23 ± 7	21.7 ± 5.5	18.7 ± 5.2
SpO_2_, (%)	96.4 ± 7.1	91.3 ± 19.1	96.9 ± 6.6	96.9 ± 3.8	92.8 ± 16.4	98.2 ± 6.1	95.8 ± 8.4	91.9 ± 19.2	98.3 ± 5.1
Body Temperature, (°C)	36.9 ± 0.7	36.5 ± 1.8	36.8 ± 0.7	36.8 ± 0.8	36.6 ± 1.2	36.7 ± 0.9	36.9 ± 0.5	36.6 ± 0.6	36.7 ± 0.8
ASA Classification, (1–6)	3.4 ± 1.1	3.6 ± 0.9	3.1 ± 1.1	4.0 ± 1.1	3.3 ± 0.8	2.7 ± 0.9	3.5 ± 0.9	4.1 ± 0.9	2.6 ± 0.9
Treatment History ^†^, *n* (%)	9 (0.6)	76 (12.5)	10 (0.5)	2 (0.6)	16 (10.3)	44 (0.5)	9 (1.2)	14 (13.7)	70 (0.3)
Operation ^‡^, *n* (%)	116 (8.4)	45 (7.4)	175 (8.7)	20 (5.7)	12 (7.8)	5768 (60.5)	66 (8.8)	8 (7.8)	11,673 (58.6)

^†^ Treatment history: any pharmacological treatment or additional oxygen supply that could affect the vital signs at the time of measurement. ^‡^ Operation: major surgery within one week of event occurrence.

**Table 4 jcm-08-01336-t004:** Acute respiratory failure prediction performance. AUROC = area under the receiver operating characteristic curve; PPV = positive predictive value; NPV = negative predictive value.

Time	Model	AUROC	Sensitivity	Specificity	PPV	NPV	Accuracy	F2-Score
**1 h**	MEWS	0.634	0.245	0.876	0.156	0.925	0.822	0.191
	NEWS	0.641	0.518	0.705	0.141	0.940	0.689	0.222
	FAST-PACE	0.886	0.830	0.777	0.259	0.980	0.782	0.394
**2 h**	MEWS	0.624	0.229	0.876	0.137	0.930	0.825	0.171
	NEWS	0.628	0.498	0.705	0.127	0.943	0.689	0.202
	FAST-PACE	0.886	0.881	0.742	0.226	0.986	0.753	0.360
**4 h**	MEWS	0.615	0.213	0.876	0.120	0.934	0.827	0.154
	NEWS	0.616	0.479	0.705	0.114	0.945	0.689	0.184
	FAST-PACE	0.868	0.771	0.800	0.234	0.978	0.798	0.359
**6 h**	MEWS	0.607	0.201	0.876	0.109	0.935	0.829	0.142
	NEWS	0.608	0.467	0.705	0.107	0.946	0.689	0.174
	FAST-PACE	0.869	0.837	0.748	0.201	0.984	0.754	0.324

**Table 5 jcm-08-01336-t005:** Cardiac arrest prediction performance.

Time	Model	AUROC	Sensitivity	Specificity	PPV	NPV	Accuracy	F2-Score
**1 h**	MEWS	0.746	0.410	0.876	0.089	0.981	0.863	0.146
	NEWS	0.759	0.702	0.705	0.066	0.988	0.705	0.120
	FAST-PACE	0.896	0.836	0.777	0.100	0.994	0.779	0.178
**2 h**	MEWS	0.745	0.406	0.876	0.085	0.981	0.863	0.140
	NEWS	0.757	0.697	0.705	0.063	0.988	0.705	0.115
	FAST-PACE	0.891	0.870	0.742	0.087	0.995	0.745	0.158
**4 h**	MEWS	0.741	0.397	0.876	0.078	0.982	0.864	0.130
	NEWS	0.753	0.691	0.705	0.058	0.989	0.705	0.107
	FAST-PACE	0.893	0.814	0.800	0.097	0.994	0.801	0.173
**6 h**	MEWS	0.737	0.388	0.876	0.075	0.982	0.864	0.125
	NEWS	0.750	0.685	0.705	0.056	0.989	0.705	0.104
	FAST-PACE	0.886	0.857	0.748	0.080	0.995	0.751	0.147

**Table 6 jcm-08-01336-t006:** Net reclassification index (NRI) in predicting acute respiratory failure.

Time	Model	NRI (Event)	NRI (No Event)	NRI
**1 h**	MEWS to FAST-PACE	0.426	−0.099	0.327
	NEWS to FAST-PACE	0.134	0.072	0.206
**2 h**	MEWS to FAST-PACE	0.464	−0.135	0.329
	NEWS to FAST-PACE	0.173	0.036	0.209
**4 h**	MEWS to FAST-PACE	0.418	−0.076	0.342
	NEWS to FAST-PACE	0.124	0.095	0.219
**6 h**	MEWS to FAST-PACE	0.469	−0.128	0.341
	NEWS to FAST-PACE	0.172	0.043	0.215

**Table 7 jcm-08-01336-t007:** NRI in predicting cardiac arrest.

Time	Model	NRI (Event)	NRI (No Event)	NRI
**1 h**	MEWS to FAST-PACE	0.585	−0.099	0.486
	NEWS to FAST-PACE	0.312	0.072	0.384
**2 h**	MEWS to FAST-PACE	0.651	−0.135	0.517
	NEWS to FAST-PACE	0.383	0.036	0.419
**4 h**	MEWS to FAST-PACE	0.558	−0.076	0.482
	NEWS to FAST-PACE	0.292	0.095	0.387
**6 h**	MEWS to FAST-PACE	0.636	−0.128	0.507
	NEWS to FAST-PACE	0.370	0.043	0.412

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
