# Peer review of "Predicting Cardiac Arrest and Respiratory Failure Using Feasible Artificial Intelligence with Simple Trajectories of Patient Data"

_jcm, 2019, doi:10.3390/jcm8091336_

Round 1
Reviewer 1 Report
The authors present a smart way to predict acute cardiac arrest and respiratory failure at different time intervals with a simple clinical trait. The rationale behind their system is that features describing heart functions are described by 7 temporal paths + 2 more features.
Since paths are temporal, the use of LSTM is smart and convincing.
Reviewer 2 Report
Thank you for submitting this interesting paper to the journal.
Firstly, please clarify ethical approval for this study.
I was a little unclear by patient flow in the study- the flowchart figure indicated 27,708 patients were included, but table three seems to indicate 3,984 were included. Please clarify.
What was the definition of acute respiratory failure? If tracheal intubation was the definition, then it would be better to state this.
You mentioned using NEWS-2- how did you determine which patients had hypercapnic respiratory failure and so determine which SpO2 scale to use?
What is the value in using AI if your predictors are pre-defined? Why not just use logistic regression?
What internal validation was done? Any bootstrapping?
I am unclear how ASA can be used in non-surgical patients- please clarify.
Please provide more detail about the inclusion of treatment information? How did this work?
There is no limitations section!
Reviewer 3 Report
Thank you for an interesting submission. I have suggestions and some questions listed in order:
1. Overall:
a. Be consistent on the word choice between "variable" and "feature". By convention, "feature" is used in a machine learning context while "variable" is used in a statistical context.
b. It is unclear whether or not the predictive model only predicts the occurrence of an event (defined as intubation or cardiac arrest) or predicts the occurrence as well as the type of event.
2. Abstract:
a. Line 22 should have the features chosen explicitly listed.
3. Introduction:
a. Line 51: "A higher score in this tool is statistically..." If using the word "statistically" please shortly describe statistical methods that were cited. Otherwise, the manuscript should have a different adjective.
b. Lines 61-65 is a run-on sentence. Please re-organize the sentence "Recent studies..."
c. Line 65 describes the use of multiple values from laboratory tests or medical devices in a model that outperforms logistic regression predictions or clinical scores. Include a statement after the last sentence the reason why you choose to remove values from laboratory tests.
d. Line 66: "non-sequential" did you mean categorical?
e. Line 68: “till date” should be “to date”
4. Methods:
a. Lines 101-104 is a run-on sentence
b. Line 109: it is unclear which subset of subjects “ratio of positive to negative samples is maintained” refers to.
c. Line 111: The manuscript needs further explicit underfitting/overfitting validation
d. Figure 1: “Event” or “Acute respiratore failure” and “Cardiac arrest” needs to be consistent. The two events were stratified before splitting the dataset into training an test – why?
e. Line 117: Features should be listed here
f. Line 118: It is unclear whether your predictive model classifies two variables (0 no event, or 1 event) or three variables (0 no event, 1 acute respiratory failure, 2 cardiac arrest)
g. Line 126: Describe the feature “treatment history” and “recent history of operations” How were they encoded? What is the exact timeline and definition of “recent”?
h. Line 129: It is unclear what “refine the mixed value” means
i. Line 130: What criteria was used to determine outlier removal? What does “invalid data” mean?
j. Line 131: The manuscript needs to include the percent of data missing
k. Figure 2: ASA needs to be defined in figure 2 caption.
l. Line 138: How far back in the past do you go for treatment history? How was it encoded for the model? What kind of medications were included in the model?
m. Line 140: Was an hourly sample rate a limitation in the /study?
n. Line 151: Typo. “Class 6”
o. Line 152: “As the ASA record…” the indentation needs to be removed
p. Line 152-157: This is the weak link in the manuscript. Why use a feature that was extracted for only 10% of the subject? Furthermore, this value is nonexistent for non-surgical patients and then turned from categorical into quantitative. The inclusion of ASA as a feature in this predictive model has not been sufficiently justified.
q. Table 1. The unit of each feature should be included in the table. Also, the median should be included to give context to the range.
r. Line 165: The major differences between MEWS, NEWS, and FAST-PACE should be described.
s. Table 2. AVPU should be described in the Table caption
t. Line 176: xi needs to be properly formatted
u. Line 185: Software/programming language(s) used with version need to be included and cited in this section. Libraries used also need to be included and cited.
5. Results:
a. Table 3. Table needs to include Ethnicity and Race. Treatment History, Operation, and Emergency Operation needs to be defined. The asterisks in Age need to be defined. The n used for MEWS and NEWS scoring is unclear – the rationale behind comparing the large n to the smaller n’s of FAST-PACE is unclear.
b. Figure 4. It is unclear if this histogram is based on the whole subject database. Furthermore, your histogram shows that the highest frequency of events occur at Hour 1 and Hour 2. Your predictive model needs at least 6 hours of data to make an accurate prediction. This discrepancy makes the model lose its clinical relevancy. Please clarify.
c. Line 204. What feature or variable is the cut-off value used for?
d. Line 209: Typo “of0.607”.
e. Line 212: The word “significant” is used but no statistical tests are described.
6. Discussion
a. Line 246-249: “The main reasons for this performance difference …” Evidence to support this sentence is needed.
b. Lines 267-269 need citations for each predictive scoring system
c. Line 277 – need consistency between the use of “AI” vs “artificial intelligence”
d. Line 290: “Cost-sensitive methods” such as?
Round 2
Reviewer 3 Report
The changes look good, however I still don't see the following in the manuscript: "We trained the model in TensorFlow-GPU 1.6 [29] with Python 3.5, pandas 0.19, NumPy 1.12, and SciPy 1.01 libraries."
Also there is a typo on line 104, there should be a space after the period.
Excellent work.
Author Response
The changes look good, however I still don't see the following in the manuscript: "We trained the model in TensorFlow-GPU 1.6 [29] with Python 3.5, pandas 0.19, NumPy 1.12, and SciPy 1.01 libraries."
Also there is a typo on line 104, there should be a space after the period.
Thank you for your careful review.
I apologize for typing errors.
We have revised the script according to your comment, and marked the corrections in red.
The revised script will be sent as an attachment so please review again.